# Clinical Significance of Stem Cell Biomarkers EpCAM, LGR5 and LGR4 mRNA Levels in Lymph Nodes of Colon Cancer Patients

**DOI:** 10.3390/ijms23010403

**Published:** 2021-12-30

**Authors:** Manar AbdelMageed, Hager Tarek H. Ismail, Lina Olsson, Gudrun Lindmark, Marie-Louise Hammarström, Sten Hammarström, Basel Sitohy

**Affiliations:** 1Department of Clinical Microbiology, Infection and Immunology, Umeå University, SE-90185 Umeå, Sweden; mamohammad@zu.edu.eg (M.A.); hager_vet@hotmail.com (H.T.H.I.); lina.olsson@umu.se (L.O.); marie-louise.hammarstrom@umu.se (M.-L.H.); sten.hammarstrom@umu.se (S.H.); 2Department of Radiation Sciences, Oncology, Umeå University, SE-90185 Umeå, Sweden; 3Department of Pathology, Faculty of Veterinary Medicine, Zagazig University, Zagazig 44511, Egypt; 4Department of Clinical Pathology, Faculty of Veterinary Medicine, Zagazig University, Zagazig 44511, Egypt; 5Institution of Clinical Sciences, Lund University, SE-25187 Helsingborg, Sweden; lindmarkgudrun@gmail.com

**Keywords:** colon cancer, EpCAM, LGR5, LGR4, CEA, CXCL17, CXCL16, qRT-PCR, stem cell markers, regional lymph nodes, prognosis

## Abstract

The significance of cancer stem cells (CSCs) in initiation and progression of colon cancer (CC) has been established. In this study, we investigated the utility of measuring mRNA expression levels of CSC markers EpCAM, LGR5 and LGR4 for predicting survival outcome in surgically treated CC patients. Expression levels were determined in 5 CC cell lines, 66 primary CC tumors and 382 regional lymph nodes of 121 CC patients. Prognostic relevance was determined using Kaplan-Meier survival and Cox regression analyses. CC patients with lymph nodes expressing high levels of EpCAM, LGR5 or LGR4 (higher than a clinical cutoff of 0.07, 0.06 and 2.558 mRNA copies/18S rRNA unit, respectively) had a decreased mean survival time of 32 months for EpCAM and 42 months for both LGR5 and LGR4 at a 12-year follow-up (*p* = 0.022, *p* = 0.005 and *p* = 0.011, respectively). Additional patients at risk for recurrence were detected when LGR5 was combined with the biomarkers CXCL17 or CEA plus CXCL16. In conclusion, the study underscores LGR5 as a particularly useful prognostic biomarker and illustrates the strength of combining biomarkers detecting different subpopulations of cancer cells and/or cells in the tumor microenvironment for predicting recurrence.

## 1. Introduction

Cancer stem cells (CSCs) constitute a subpopulation of cancer cells with self-renewal and multi-lineage differentiation capabilities [1]. The crucial roles played by CSCs in tumor initiation, progression, metastasis, drug and radiation resistance and subsequent tumor recurrence are supported by an accumulating body of evidence. CSC associated with different malignancies can be identified through their expression of specific biomarkers. A number of CSC markers linked to colorectal cancer (CRC) have been described, including, but not restricted to, epithelial cell adhesion molecule (EpCAM), and leucine-rich repeat-containing G protein-coupled receptor 5 (LGR5) [2,3].

EpCAM is a transmembrane protein expressed by normal epithelial cells and cancers of epithelial origin. In addition to its role in intercellular adhesion, EpCAM also functions in cell signaling, differentiation, proliferation and migration [4]. EpCAM overexpression correlates with poor survival in some cancer types and better survival in others [5]. In CRC, the capability of cancer cells isolated from primary tumors to form xenografts when injected in immunodeficient mice was reserved to a small subset of cells showing EpCAM^high^/CD44^+^ expression [6]. In addition, EpCAM^high^/CD44^+^ expression was positively correlated with tumor invasion and metastasis in CRC patients [7].

LGR5, also known as GPR49, is a G-protein-coupled receptor expressed by normal stem cells in various tissues, including small and large intestine, where its expression is confined to the crypt base columnar cells [8]. In CRC, LGR5 has been identified as a CSC marker, whose overexpression was associated with lymph node expression, distant metastases, and reduced overall and disease-free survival [9,10,11,12,13]. LGR4 and LGR6 are close homologues to LGR5. LGR4 was also reported to enhance invasion by CC cells and metastasis, and expression levels of LGR4 was correlated to poor prognosis in CRC patients [14,15].

Recently, we discovered that chemokines CXCL17 and CXCL16 are ectopically expressed in colon cancer (CC) and could serve as biomarkers for poor prognosis [16,17]. The prognostic value of CXCL17 mRNA was even more pronounced when the analysis was restricted to tumor cells expressing low CEA mRNA levels [16,18]. Interestingly, the myeloid biomarker G-protein-coupled-receptor EMR1 was also ectopically expressed in colon cancer correlating with CXCL17, mainly detecting the same tumor cells [19]. Additionally, we found that G protein-coupled receptor 35 (GPR35) is a biomarker for poor prognosis in CC, probably also identifying less differentiated CC cells [20].

In this study, we investigated the prognostic value of measuring the mRNA expression levels of the CSC markers EpCAM, LGR5 and LGR4 in primary tumors and regional lymph nodes of CC patients. We found that elevated mRNA levels in lymph nodes of all three biomarkers predict shortened disease-free survival and that combining determinations of other CC prognostic markers including carcinoembryonic antigen (CEA), and the chemokines CXCL16 and CXCL17 with LGR5 measurements enhance its predictive ability.

## 2. Results

### 2.1. Levels of EpCAM, LGR5 and LGR4 mRNA in Primary Colon Cancer Tumors and Colon Cancer Cell Lines

The mRNA levels of EpCAM and LGR5 but not LGR4 were significantly higher in primary tumors than in normal colon tissues. The median mRNA expression level of EpCAM was two times higher in primary tumors than in normal colon tissues (170 and 82 mRNA copies per 18S rRNA unit, respectively, *p* = 0.0002). For LGR5, the median mRNA expression level was seven times higher in primary tumors than in normal colon tissues (3.8 and 0.5 mRNA copies per 18S rRNA unit, respectively, *p* < 0.0001). There was no significant difference in the median of LGR4 mRNA levels between primary tumors and normal colon tissues (18.5 and 21 mRNA copies per 18S rRNA unit, respectively, *p* = 0.8) (Figure 1A). Interestingly, when the actual values for the three biomarkers were compared pairwise in primary tumors, it was found that EpCAM and LGR4 was highly correlated (r = 0.60, *p* < 0.0001), while LGR5 showed poor correlation with the other two markers (r = 0.31, *p* = 0.013 and r = 0.32, *p* = 0.009 for EpCAM and LGR4, respectively), indicating that LGR5 at least partly identifies a different population of cells than the other two markers.

Quantification of EpCAM, LGR5 and LGR4 mRNA expression levels in a panel of five different human CC cell lines (HCT8, HT29, LS174T, Caco2 and T84) revealed that EpCAM was expressed at the highest levels, followed by LGR4, while LGR5 was expressed at the lowest levels. EpCAM and LGR5 were not detected in four human immune cell lines (Jurkat, CNB6, LGR4 and U937), while LGR4 was only detected at trace levels (data not shown). EpCAM was expressed only at trace levels in a human endothelial cell line and in foreskin fibroblasts in comparison to the CC cell lines. The situation was different for LGR5, which showed no expression in endothelial cells but had comparatively high levels in fibroblasts. LGR4 was expressed at relatively high levels in both of these cell types (Figure 1A).

### 2.2. Levels of EpCAM, LGR5 and LGR4 mRNA in Regional Lymph Nodes of Colon Cancer Patients

The mRNA expression levels of EpCAM, LGR5 and LGR4 were assessed in a panel of 382 regional lymph nodes from 121 CC patients representing all four TNM-stages and 77 lymph nodes from 13 patients with noncancerous disease. Average expression levels of all three mRNAs increased with TNM-stage. Median expression levels of EpCAM mRNA were 0.024, 0.023, 0.032 and 0.16 mRNA copies/18S rRNA unit in TNM-stages I, II, III and IV, respectively. Expression levels were significantly higher in lymph nodes of stage III patients compared to those of stage II patients (*p* = 0.03) and in lymph nodes of stage IV patients compared to those of stage I (*p* < 0.0001), stage II (*p* < 0.0001) and stage III (*p* = 0.02) patients (Figure 1B). Median expression levels of LGR5 mRNA were 0.011, 0.012, 0.018 and 0.062 mRNA copies/18S rRNA unit in TNM-stages I, II, III and IV, respectively. Expression levels were significantly higher in lymph nodes of stage IV patients compared to those of stage I (*p* = 0.007) and stage II (*p* = 0.002) patients (Figure 1C). Median expression levels of LGR4 mRNA were 0.60, 0.58, 0.70 and 2.5 mRNA copies/18S rRNA unit in TNM-stages I, II, III and IV, respectively. Expression levels were significantly higher in lymph nodes of stage IV patients compared to those of stage I (*p* < 0.0001), stage II (*p* < 0.0001) and stage III (*p* = 0.003) patients (Figure 1D). All three biomarkers were readily detected in control lymph nodes (Figure 1B–D).

Based on histopathological examination, the 382 lymph nodes were differentiated into H&E(+) where cancer cells were observed (*n* = 22) and H&E(-) where they were not detected (*n* = 360). The mRNA expression levels of EpCAM, LGR5 and LGR4 were significantly higher in H&E(+) than in H&E(-) lymph nodes. The median values of EpCAM mRNA were 3.5 × 10^3^ times higher in H&E(+) than H&E(-) lymph nodes (82.9 and 0.02 mRNA copies/18S rRNA unit, respectively, *p* < 0.0001) (Figure 2A). The median values of LGR5 mRNA were 30 times higher in H&E(+) than in H&E(-) lymph nodes (0.38 and 0.013 mRNA copies/18S rRNA unit, respectively, *p* < 0.0001) (Figure 2C). The median values of LGR4 mRNA were nine times higher in H&E(+) than H&E(-) lymph nodes (5.1 and 0.60 mRNA copies/18S rRNA unit, respectively, *p* < 0.0001) (Figure 2E).

To provide additional evidence that high expression levels of these biomarker mRNAs are related to presence of CC tumor cells, the lymph nodes were divided into three groups based on their previously determined CEA mRNA expression levels: CEA(+), CEA(int) and CEA(-). CEA mRNA levels in the first group is above the clinical cut-off (>3.67 CEA mRNA copies/18S rRNA unit), between 0.013–3.67 and <0.013 mRNA copies/18S rRNA unit in the second and third group, respectively [21]. All three markers showed significantly higher mRNA levels in the CEA(+) group compared to the CEA(int) and CEA(-) groups (*p* < 0.0001). The median levels of EpCAM mRNA were 72.5, 0.04 and 0.02 mRNA copies/18S rRNA units in CEA(+), CEA(int) and CEA(-) lymph nodes, respectively (Figure 2B). The median levels of LGR5 mRNA recorded were 1.13, 0.014 and 0.011 mRNA copies/18S rRNA units in CEA(+), CEA(int) and CEA(-) lymph nodes, respectively (Figure 2D), while those of LGR4 mRNA were 7.50, 0.61 and 0.59 mRNA copies/18S rRNA units in CEA(+), CEA(int) and CEA(-) lymph nodes, respectively (Figure 2F).

### 2.3. Correlations between mRNA Expression Levels of EpCAM, LGR5, LGR4, CEA, CXCL17, CXCL16 and GPR35 V2/3 in Regional Lymph Nodes of Colon Cancer Patients

The mRNA expression levels of CEA, CXCL17, CXCL16 and GPR35 V2/3 have been previously determined in the same 382 lymph nodes studied in this work [17,18,20,21]. We investigated whether the three CSC biomarkers would group together with any or all of the previously investigated biomarkers. The three CSC markers grouped together with CXCL16 and GPR35V2/3 and with each other as demonstrated by very high *p*-values and high r-values (Table 1). In contrast, CEA and CXCL17 did not show high *p*-values or r-values in stages I and II when compared to the CSC markers (Table 1).

### 2.4. Colocalization of LGR5, CEA and EpCAM Proteins in Primary Colon Cancer Tumors

To investigate where the LGR5 and CEA proteins are localized compared to the EpCAM protein in primary tumors, a two-color immunofluorescence experiment was performed using anti-LGR5 and anti-CEA antibodies. The sections were then double-stained with the anti-EpCAM mAb BerEP4 antibody. The merged photomicrographs presented in (Figure 3C,G) demonstrate that LGR5 colocalizes with EpCAM all over the epithelial cells, while CEA and EpCAM colocalize only at the luminal surface of the epithelium. 

### 2.5. Clinical Relevance of EpCAM, LGR5 and LGR4 mRNA Expression Levels in Lymph Nodes for Predicting Colon Cancer Recurrence after Surgery

The relevance of high expression levels of EpCAM, LGR5 and LGR4 mRNA in regional lymph nodes of CC patients for prediction of disease recurrence and influence of survival time after surgery was determined by calculating hazard risk ratio using Cox regression analysis and Kaplan-Meier survival model combined with the log-rank test. Each patient was represented by the lymph node with the highest expression level, and a cut-off level discriminating between patients with high and low risk for recurrence was determined for each marker. A summary of these survival analyses is shown in Table 2 and Table 3.

For EpCAM, the patients were divided into two groups according to the median of the expression level in the highest lymph nodes of the CC patients in stages III and IV, which was 0.07 mRNA copies/18S rRNA unit, corresponding to the 67th percentile. Patients in the high expression group [EpCAM(+) group, *n* = 40] showed a 2.5-fold increased recurrence rate compared to the low expression group [EpCAM(-) group, *n* = 81] when followed for five years and 2.1-fold at a follow-up time of 12 years (*p* = 0.010 and *p* = 0.025, respectively). A difference in mean survival times amounting to 8 months in 5 years and 34 months in 12 years after surgery (*p* = 0.007 and *p* = 0.022, respectively) were observed based on Kaplan-Meier survival analysis (Figure 4A). When the analysis was restricted to CC patients with CEA levels above the control level, the recurrence rate was 2.2-fold higher. However, this difference was not statistically significant either at 5 years or 12 years follow-up time (*p* = 0.062 and *p* = 0.063, respectively). The recurrence was associated with decreased mean survival time by seven months in five years and by 36 months in 12 years (*p* = 0.055, *p* = 0.056, respectively) when compared to the EpCAM(-) group (Figure 4B). Therefore, restricting the analysis to patients with positive CEA mRNA values does not add prognostic information compared to analyzing the entire patient group for EpCAM.

For LGR5, the patients were divided into two groups according to the median of the expression level (0.06 mRNA copies/18S rRNA unit) in all stage IV lymph nodes, corresponding to the 70th percentile. Patients in the high expression group [LGR5(+) group, *n* = 38] showed a 3-fold increased recurrence rate compared to the low expression group [LGR5(-) group, *n* = 83] when followed for 5 years and 2.5-fold when followed for 12 years (*p* = 0.002 and *p* = 0.007, respectively). The associated decrease in mean survival time was 8 and 42 months in 5 and 12 years after surgery (*p* = 0.001 and *p* = 0.005, respectively), according to Kaplan-Meier analysis (Figure 4C). When the analysis was restricted to CC patients with CEA levels above the control level, the recurrence rate was a 2.8-fold higher in the LGR5(+) group when followed for 5 years and 3-fold when followed for up to 12 years (*p* = 0.012 and *p* = 0.008, respectively) with decreased mean survival time by 11 months in five years and by 50 months in 12 years (*p* = 0.009, *p* = 0.006, respectively) when compared to the LGR5(-) group (Figure 4D). Thus, restricting the analysis to patients with positive CEA mRNA levels increased the prognostic value of the LGR5 analysis. 

For LGR4, the clinical cut-off used to divide the patients into two groups was the 75th percentile of LGR4 mRNA expression levels in the highest lymph nodes from the entire group of CC-patients i.e., 2.558 mRNA copies/18S rRNA units. Patients in the high expression group [LGR4(+) group, *n* = 30] showed a 2.8-fold increased recurrence rate compared to the low expression group [LGR4(-) group, *n* = 91] when followed for five years against 2.3-fold at 12 years follow-up time (*p* = 0.004 and *p* = 0.014, respectively). This was coupled with a decreased mean survival time of 11 and 42 months after 5 and 12 years from surgery (*p* = 0.002 and *p* = 0.011, respectively), based on Kaplan-Meier analysis (Figure 4E). The analysis was then restricted to include only the CC patients with CEA levels above the control level, resulting in 2.4- and 2.5-fold higher recurrence in the LGR4(+) group when followed for 5 and 12 years, respectively (*p* = 0.029 and *p* = 0.024, successively). This was coupled with 9- and 46-months decreased survival times in 5 and 12 years, respectively (*p* = 0.024, *p* = 0.019, successively) when compared to the LGR4(-) group (Figure 4F), demonstrating a modest increase in the prognostic value of LGR4 clinical cut-off at 12 years follow up when considering CEA mRNA values. 

Given the observed advantage of combining LGR5 analysis with CEA analysis as compared to analysis of LGR5 alone (Figure 4C,D), we investigated whether increased survival time between LGR5(+) and LGR5(-) patients could be achieved if LGR5 was combined with other biomarkers or restricted to patients of a certain TNM stage. Figure 4G–J shows some results of these analyses. When restricted to patients in TNM-stage I only, the difference in the recurrence rate increased to 13.3-fold in the high expression group [LGR5(+) group, *n* = 4] compared to the low expression group [LGR5(-) group, *n* = 19] when patients were followed for 5 years against 8.7-fold when followed for 12 years (*p* = 0.036 and *p* = 0.077, respectively). The mean survival time difference increased to 18 and 54 months in five and twelve years after surgery, respectively (*p* = 0.007 and *p* = 0.033) (Figure 4G). 

Confining the analysis to only patients in the CEA(int) and the CXCL16(+) groups (CXCL16 mRNA values above the clinical cut-off; >7.2 CXCL16 mRNA copies/18S rRNA unit) revealed that the patients in the group with high LGR5 levels [LGR5(+) group, *n* = 6] had 7.6- and 10.4-fold increased recurrence rates compared to the low expression group [LGR5(-) group, *n* = 13] when followed for 5 and 12 years after surgery, respectively (*p* = 0.079 and *p* = 0.044, respectively). These changes coincided with decreased mean survival times of 11 and 63 months (*p* = 0.039 and *p* = 0.014, respectively), according to Kaplan-Meier analysis (Figure 4H).

Figure 4I shows the result of dividing CXCL17(+) CC patients (CXCL17 mRNA values higher than 0.0012 mRNA copies/18S rRNA unit; 73rd percentile) into a LGR5(+) and a LGR5(-) group. The LGR5(+) group, [*n* = 22] had 6.1 and 6.2-fold increased recurrence rates compared to the low expression group [*n* = 10] when followed for 5 and 12 years after surgery (*p* = 0.087 and *p* = 0.082, respectively). The LGR5(+) group had decreased mean survival times of 14 and 51 months (*p* = 0.050 and *p* = 0.047, respectively).

Figure 4J, finally, shows the result of restricting the analysis to patients that have CEA levels above the control level and also have high CXCL17 levels. Patients in the high expression group [LGR5(+) group, *n* = 19] had a 5.8-fold and 6.8-fold increased recurrence rate compared to the low expression group [LGR5(-) group, *n* = 9] when followed for 5 and 12 years after surgery, respectively (*p* = 0.096 and *p* = 0.075, respectively). Associated decreased mean survival times were 14 and 53 months (*p* = 0.060 and *p* = 0.043, respectively). 

Combining EpCAM and LGR4 clinical cut-offs with other CC prognostic markers did not improve the discriminating power between the marker positive and the marker negative groups (data not shown).

### 2.6. Absence of Correlation between the Risk of Recurrence and Survival Time after Surgery with the Levels of EpCAM, LGR5 and LGR4 mRNA Expression in Colon Cancer Primary Tumors

No difference in recurrence risk or survival time was noticed in CC patients at any of the three CSC markers EpCAM, LGR5 and LGR4 using the median mRNA of primary CC tumors as cut-off (median: 170 mRNA copies/18S rRNA unit, 3.8 mRNA copies/18S rRNA unit and 18.5 mRNA copies/18S rRNA unit, respectively).

## 3. Discussion

CRC is the second leading cause of cancer mortalities worldwide [22], closely associated with distant metastases [23]. Detection of metastatic cells in regional lymph nodes is crucial for predicting the prognosis of the disease and subsequent selection of treatment options. In CC, the current guidelines for detecting metastasis to lymph nodes are through histopathological examination of at least 12 regional lymph nodes for presence or absence of tumor cells in H&E-stained sections. The observation that about 25% of CC patients in TNM-stages I and II recur after curative surgery, although no tumor cells were detected in the regional lymph nodes, indicates that complementary methods of tumor cell detection are needed [24]. We have previously shown that measuring the mRNA expression levels of CEA, CXCL17, GPR35 and CXCL16 in regional lymph nodes of CC patients are valuable tools for predicting risk for recurrence [5,6,21,22,23]. Moreover, CXCL17 and GPR35 might specifically identify less differentiated tumor cells [16,18,20]. We have also found that the chemokine CXCL16 adds prognostic information to the classical biomarker CEA if both biomarkers are analyzed in combination [17].

This study explored the prognostic utility of measuring mRNA expression levels of three CSC biomarkers, i.e., EpCAM, LGR5 and LGR4, in primary tumors and regional lymph nodes of CC patients and the possible advantage of combining them with the abovementioned markers to further detect different subgroups of tumor cells and/or combinations of tumor cell subgroups and other cell types supporting tumor growth. CSCs may have a rate-limiting role in cancer metastasis, and the ability of CSC markers to predict disease progression and patients’ survival is being intensely investigated [1,2]. A consensus is not reached on the prognostic relevance of CSCs in CRC, which can be partly attributed to the plasticity of CSCs and the diversity in the methodology of assessment of expression. Previous studies have mainly assessed LGR5 expression by immunohistochemistry in primary tumors [12]. Two research groups used in situ hybridization to detect LGR5 mRNA in primary tumor tissues and did not find correlation between LGR5 expression and poor survival [25,26]. On the other hand, Wang et al. recorded expression of LGR5 mRNA in ~60% of circulating tumor cells in CRC patients and reported a high correlation between LGR5 expression and development of metastasis [27].

Our data are in-line with the results of RNA determination of liquid biopsies and highlights the significance of mRNA analysis of regional lymph nodes to identify patients with bad prognosis. Moreover, it points to the importance of technology for tumor identification, qRT-PCR being a more accurate technique than immunohistochemistry, and lymph nodes a better study object than the primary tumor.

The prognostic value of LGR5 mRNA levels was significantly increased when combined with the measurement of CEA, CXCL17 and CXCL16 mRNA levels. When confined to patients with nodal CEA mRNA values above the control background, the difference in survival between CC patients in the high and low LGR5 groups increased to 50 months at 12 years follow-up compared to 42 months if LGR5 is considered alone. Similarly, when patients expressing high mRNA levels of CXCL17 were investigated, the high LGR5 group displayed 51 months lower mean survival time than the low LGR5 group at 12 years follow-up. Moreover, abysmal prognosis was identified in a group of patients expressing high CXCL16 and LGR5, but intermediate CEA mRNA levels, where the difference between the low and high LGR5 groups was as large as 63 months at 12 years follow-up, and three of the six patients in the LGR5(+) group were stage II patients. Thus, it is reasonable to assume that CEA and LGR5 detects partly different CC tumor cell-population, i.e., CEA detects mainly fully differentiated transformed colonocytes, and LGR5 mainly transformed undifferentiated colonocytes. Therefore, the two markers complement each other as CC markers. This perspective is supported by the modest correlation observed between LGR5 and CEA expression in lymph nodes (r = 0.28) and the results of the two-color immunofluorescence staining in the primary tumor tissues, where CEA and EpCAM colocalize only at the epithelium’s luminal surface, while LGR5 colocalizes with EpCAM all over the epithelial cells, suggesting the main expression of CEA in highly differentiated tumor cells, following earlier studies [28]. These results also agree with earlier studies demonstrating that immunohistochemistry is not an efficient method in detecting LGR5 unlike situ hybridization method, which is effective in detecting the localization of LGR5 mainly in less differentiated tumors cells [29]. Measurements of CXCL16 and CXCL17 strengthen the prognostic value of LGR5 by detecting high number of aggressive tumor cells and/or tumor growth promoting cells in the tumor cell microenvironment, as discussed earlier [17,18]. It is an important goal to identify new, high-risk groups who would benefit from adjuvant chemotherapy or other new treatments that may decrease the relapse incidence and increase cancer-free survival, such that the low-risk groups would skip adjuvant treatment associated with unnecessary side effects.

An interesting observation, also pointing to the importance of LGR5 as a biomarker for CC, was the finding that high levels of LGR5 identified a group of stage I patients with a very poor prognosis. Strikingly, restricting the analysis to stage I patients only showed a significant increase in the recurrence rate to 13.3-fold in the high expression group compared to the low expression group when patients were followed for 5 years and 8.7-fold when followed for 12 years. This led to an increase in the mean survival time difference to 18 and 54 months in five and twelve years after surgery, respectively. Because this finding in TNM stage I patients is based on a limited number of patients, further analysis of a larger number of patients is necessary to validate these results. The fact that only patients in stage I and not in stage II show notable prognostic dependence on LGR5 expression is intriguing.

## 4. Materials and Methods

### 4.1. Patients and Tissue Specimens for mRNA Analysis

Primary tumor specimens were retrieved from 66 CC patients (30 men and 36 women; median age 74 years, range 42–88 years) after surgery. None of the patients received treatment before surgery. Fourteen patients were in stage I (T1-2N0M0), 30 in stage II (T3-4N0M0), 17 in stage III (anyTN1-2M0) and 5 in stage IV (anyTanyNM1). The tumor samples were collected immediately after resection, snap-frozen and stored at −70 °C until RNA extraction. Normal colon samples retrieved from resection margins of CC tumors were collected from 30 patients (17 men and 13 women; median age 72 years, range 57–85) and treated the same way. 

Lymph nodes were collected from 121 cancer patients (55 men and 66 women; median age 73 years, range 42–89 years): 73 lymph nodes were from 23 patients in stage I, 190 nodes were from 52 patients in stage II, 88 nodes were from 37 patients in stage III, and 31 nodes were from 9 patients in stage IV. Twenty-two lymph nodes were judged positive for disseminated tumor cells by routine histopathology (H&E(+)) and 360 lymph nodes H&E(-). Control lymph nodes (*n* = 77) were from 13 patients (10 men and 3 women; median age 23 years, range 9–32 years). Eleven of the controls had ulcerative colitis, one had Crohn’s disease and one patient had lipoma.

### 4.2. Cell Lines

Five human CC cell lines (LS174T, HT29, T84, HCT8 and CaCo2), a T cell line Jurkat, two B cell lines CNB6 and KR4, a monocyte cell line U937, an endothelial cell line HUVEC and primary foreskin fibroblasts (FSU) were cultured and analyzed for mRNA expression. Culture conditions and sources are as described previously [30,31].

### 4.3. Real-Time qRT-PCR

For absolute quantification of EpCAM, LGR5 and LGR4 mRNA in lymph nodes, we constructed real time qRT-PCR assays using specific primers placed in different exons and a reporter dye-labeled probe hybridizing over the exon boundary in the amplicon and specific RNA copy standards for the quantification. For EpCAM mRNA (NM_002354.3), the primers and probe sequences were forward primer 5′-CAGTTGGTGCACAAAATACTGTCA-3′, reverse primer 5′-TTCTGCCTTCATCACCAAACA-3′, and probe 5′-CTCAAAGCTGGCT-3′. For LGR5 mRNA, the assay detects the four transcript variants (NM_003667.4, NM_001277226.2, NM_001277227.2, NR_110596.2). The primers and probe sequences were forward primer 5′-CCTTCATTCAGTGCAGTGTTCAC-3′, reverse primer 5′-TCAGCCAGCCATCAAGCA-3′, and probe 5′-TTCCCCAGGCCCCTT-3′. For LGR4 mRNA, the assay detects both transcript variants (NM_018490.5, NM_001346432.2). The primers and probe sequences were forward primer 5′-AGCCATTCGAGGGCTGAGT-3′, reverse primer 5′-ACTGAGGTAATATGGTTGGCATCTAA-3′ and probe 5′-CTTTGCAGTCTTTGCG-3′. The reporter dye was FAM and the quencher dye was NFQ-MGB. The size of the amplicon was 68 bp for EpCAM, 74 bp for LGR5 and 63 bp for LGR4. The qRT-PCR profile was 60 °C for 5 min and 95 °C for 1 min, which is followed by 45 cycles of 95 °C for 15 s and 60 °C for 1 min. RNA oligonucleotides with sequences identical to those in the areas amplified in the qRT-PCR assays were custom synthesized at Dharmacon (Lafayette, CO, USA). Serial dilutions of the RNA copy standards at concentrations from 10^3^ to 10^8^ copies per µL were included in each qRT-PCR run. Concentrations in unknown samples were determined from the standard curve and expressed as copies of mRNA per µL. The concentration of 18S rRNA was expressed as arbitrary units from a standard curve of serial dilutions of a preparation of total RNA from human peripheral blood mononuclear cells. One unit was defined as the amount of 18S rRNA in 10 pg RNA [16]. EpCAM, LGR5 and LGR4 mRNA were expressed as copies per unit of 18S rRNA. Real time qRT-PCR assays for CEA, CXCL17, CXCL16 and GPR35 V2/3 mRNAs were described earlier [17,18,20,21].

### 4.4. Two-Color Immunofluorescence

Primary tumor tissue sections were cut and fixed as described previously [32]. Subsequently, the sections were incubated with primary unconjugated anti-LGR5 mAb (mouse IgG1, clone OTI2A2, MA5-25644, Thermo Fisher, Waltham, MA, USA) or anti-CEA mAb (mouse IgG1, clone II-7, M7072, Dako, Glostrup, Denmark) followed by the secondary antibody Alexa Fluor 549-conjugated goat anti-mouse IgG (red) (ab150116, Abcam, Cambridge, MA, USA). Afterwards, the sections were incubated with FITC-conjugated BerEP4 (green) (mouse IgG1, clone BerEP4, F0860; Dako). Double-positive cells show a yellow-orange color, as previously described [33]. Mouse IgG, ready to use (Dako) and FITC-conjugated anti-mouse (F0313; Dako) were used as negative controls. The sections were finally mounted with SlowFade^®^ Gold Antifade Mountant (Thermo Fisher). Microscopy was done using a Nikon fluorescence microscope and images were analyzed with NIS elements software.

### 4.5. Statistical Analysis

The statistical significance of differences in mRNA levels in primary CC tumors compared to normal colon tissues, in H&E(+) compared to H&E(-) lymph nodes, and between sets of lymph nodes with different CEA levels were calculated using the two-tailed Mann-Whitney rank sum test. Statistical significance of differences in mRNA levels between control lymph nodes and lymph nodes from different patient groups were analyzed using Kruskal-Wallis one-way analysis of variance (ANOVA) test followed by the Dunn’s multiple comparison post hoc test. Correlations between different mRNA levels were analyzed using the nonparametric Spearman correlation coefficient. The software utilized for statistical calculations was GraphPad Prism 6 (Graphpad Software, San Diego, CA, USA).

The SPSS software (IBM Corporation, Armonk, NY, USA) was used for statistical analyses of differences between patient groups in disease-free survival time and analyses of risk for recurrent disease after surgery, according to the Kaplan-Meier survival model in combination with the log-rank test and univariate Cox regression analysis. A *p*-value ≤ 0.05 was considered to be statistically significant.

### 4.6. Ethical Considerations

All procedures performed in studies involving human participants were in accordance with the ethical standards of the institutional research committee and with the 1964 Helsinki Declaration and its later amendments or comparable ethical standards. Tumor samples and lymph nodes were collected after patients’ written, informed consent. The study was approved by the Local Ethics Research Committee of the Medical Faculty, Umeå University, Umeå, Sweden (Registration number: 03-503; date of approval: 3 December 2003).

## 5. Conclusions

In conclusion, this study shows that overexpression of the CSC markers EpCAM, LGR5 and LGR4 mRNA in regional lymph nodes correlates with poor prognosis in CC patients. Each of the markers is independently useful in predicting disease outcome. LGR5 is the preferred marker because it identifies immature cancer cells in a more specific way than the other two biomarkers. Further prognostic information is obtained if LGR5 determination is combined with determination of the established biomarker CEA and the chemokines CXCL16 and CXCL17.

## Figures and Tables

**Figure 1 ijms-23-00403-f001:**
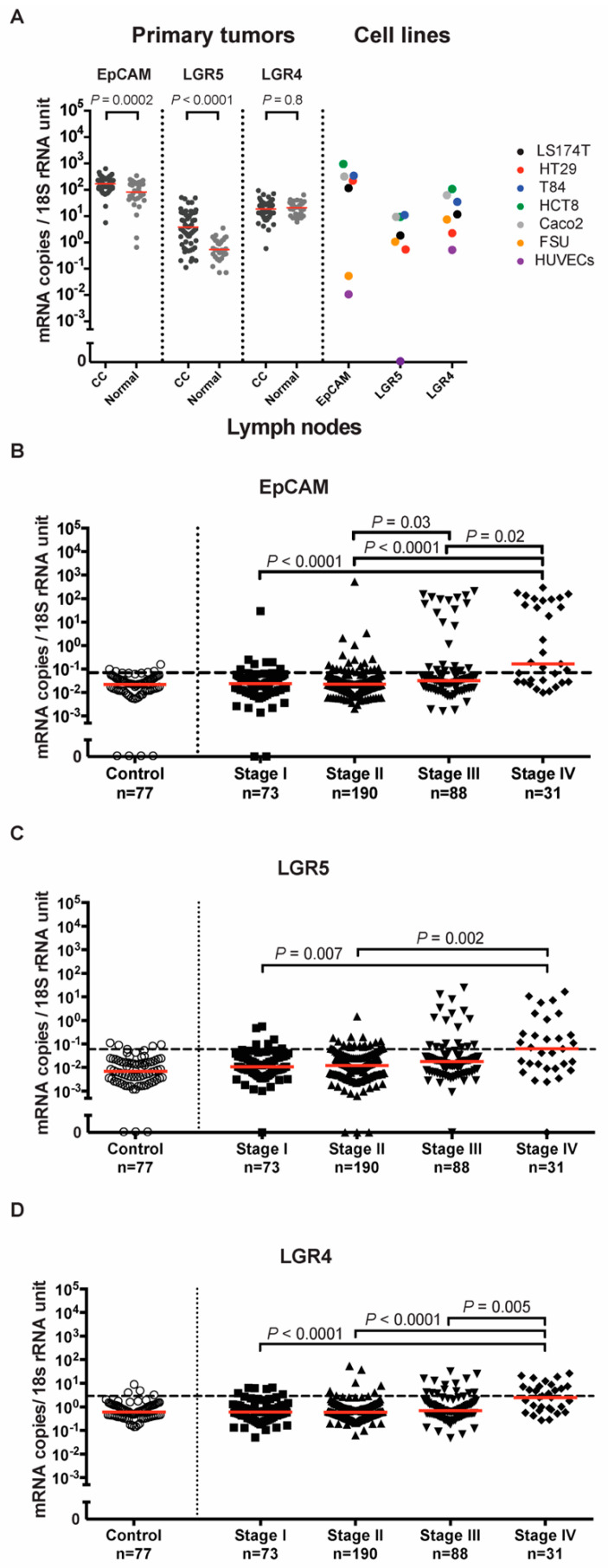
EpCAM, LGR5, and LGR4 mRNA expression levels in tissues and cell lines. (**A**) mRNA expression levels in primary colon cancer (CC) tissues, resected normal colon tissues, and in a panel of colon cancer cell lines; LS174T, HT29, T84, HCT8 and Caco2, primary foreskin fibroblast cells (FSU) and endothelial cells (HUVECs). Red horizontal lines indicate median values. (**B**–**D**) mRNA expression levels in lymph nodes from noncancerous disease patients (Control) and colon cancer patients in different TNM stages (Stage I–IV). Red horizontal lines indicate median values. Dashed horizontal lines indicate clinical cutoff values of 0.07, 0.06 and 2.558 mRNA copies/18S rRNA unit for EpCAM, LGR5 and LGR4, respectively. n = number of lymph nodes. *p*-values were calculated by two-tailed Mann-Whitney test in (**A**) and Kruskal-Wallis nonparametric ANOVA followed by post hoc Dunn’s test for multiple comparisons in (**B**–**D**).

**Figure 2 ijms-23-00403-f002:**
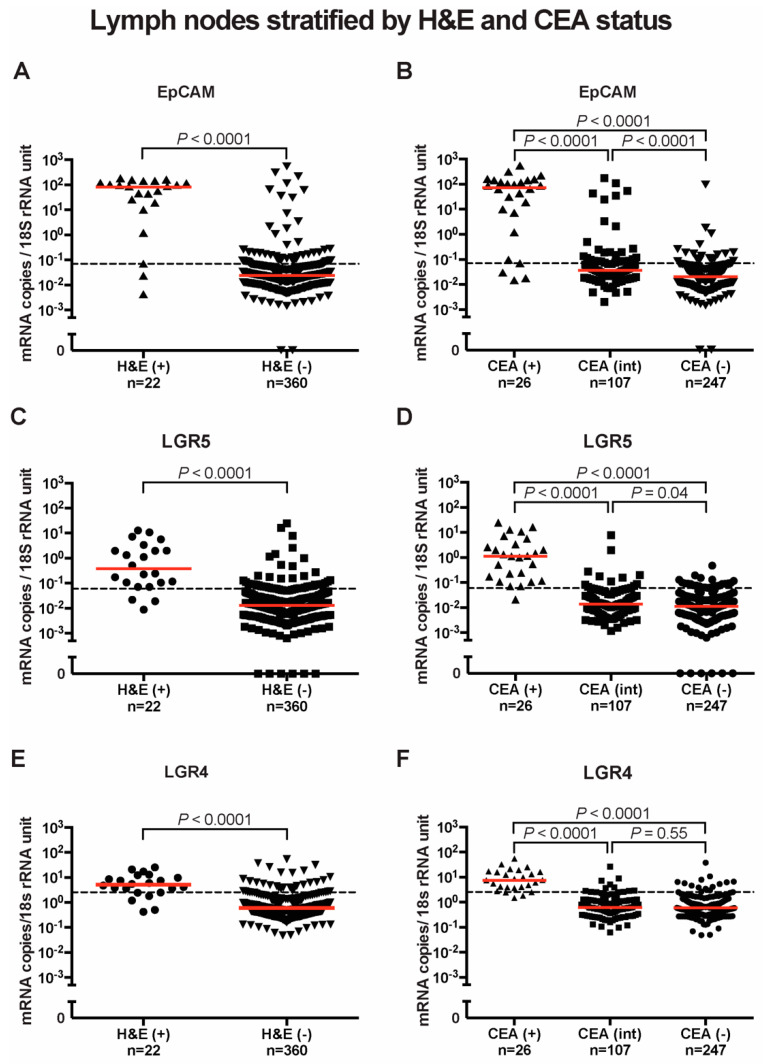
EpCAM, LGR5 and LGR4 mRNA expression in lymph nodes stratified by H&E and CEA status. (**A**) EpCAM, (**C**) LGR5, and (**E**) LGR4 mRNA levels in nonmetastatic (H&E(-)) and metastatic (H&E(+)) lymph nodes. In (**B**,**D**,**F**), lymph nodes were divided into three groups according to their CEA mRNA levels; CEA(-) = CEA mRNA levels < 0.013 copies/18S rRNA unit, CEA(int) = intermediate CEA mRNA levels, that is 0.013-3.67 copies/18S rRNA unit, and CEA(+) = CEA mRNA levels > 3.67 copies/18S rRNA unit. Red horizontal lines indicate median values. Dashed horizontal lines indicate clinical cutoff values of 0.07, 0.06 and 2.558 mRNA copies/18S rRNA unit for EpCAM, LGR5 and LGR4 respectively. n = number of lymph nodes. *p*-values were calculated by two-tailed Mann-Whitney test for comparison between expression levels in (**A**,**C**,**E**) and by Kruskal-Wallis nonparametric ANOVA followed by post hoc Dunn’s test for multiple comparisons in (**B**,**D**,**F**).

**Figure 3 ijms-23-00403-f003:**
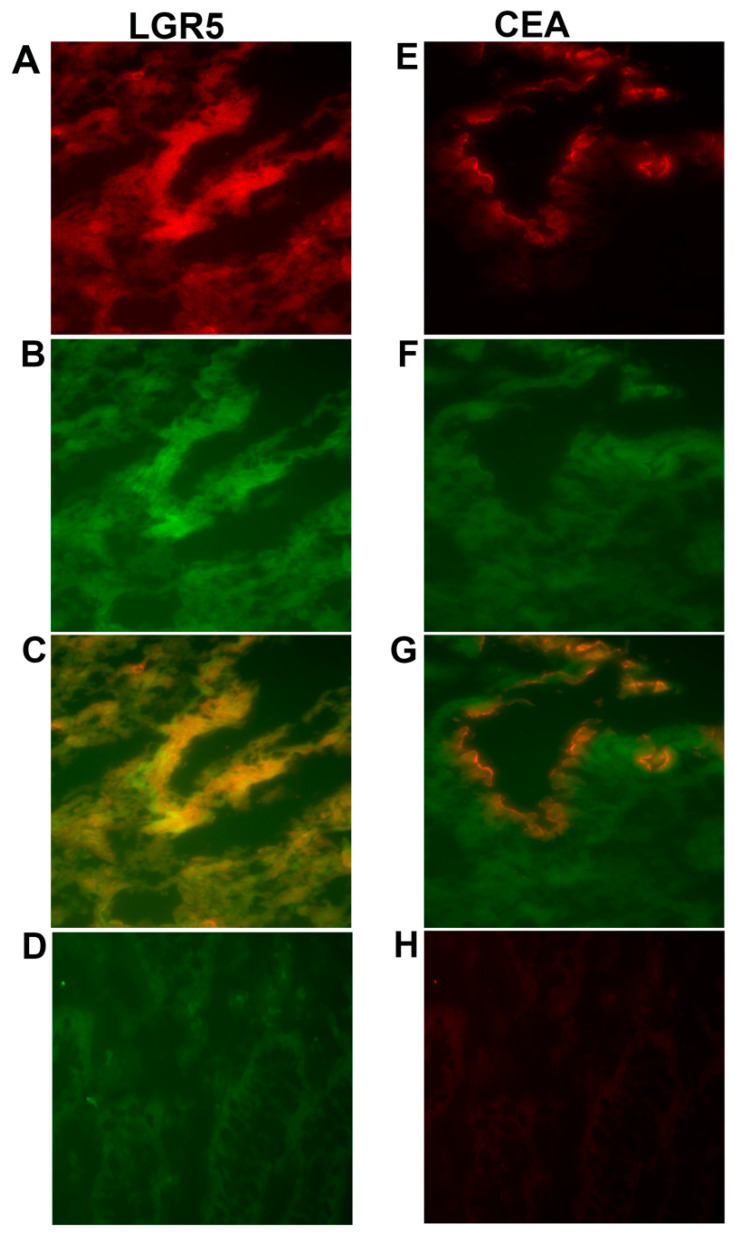
Two-color immunofluorescence staining of primary colon cancer tissue with anti-LGR5 and BerEP4, and anti-CEA and BerEP4. (**A**) Anti-LGR5, (**E**) Anti-CEA both red color. (**B**,**F**) BerEP4 mAb, green color. (**C**,**G**) Overlays giving yellow color of double-stained areas. (**D**) FITC-conjugated mouse IgG; negative control for BerEP4. (**H**) Rabbit IgG; negative control for anti-LGR5 and anti-CEA. Original magnification: ×200.

**Figure 4 ijms-23-00403-f004:**
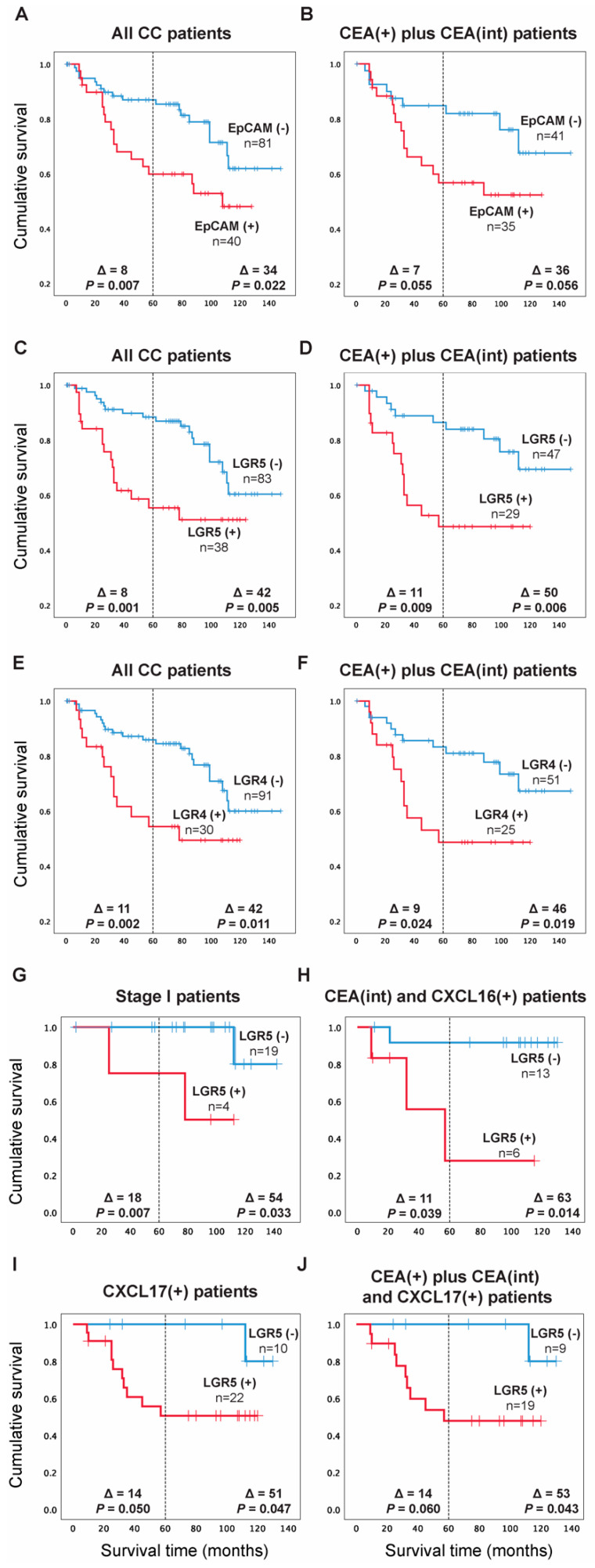
Kaplan-Meier cumulative survival curves for CC patients divided into two groups in (**A**) EpCAM(-) and EpCAM(+) according to the median of the expression level in the highest lymph nodes of the CC patients in TNM stages III and IV (0.07 mRNA copies/18S rRNA unit). (**C**) LGR5(-) and LGR5(+) according to the median of the expression level in lymph nodes of all CC patients in TNM stage IV (0.06 mRNA copies/18S rRNA unit). (**E**) LGR4(-) and LGR4(+) according to the 75th percentile of LGR4 mRNA expression values in all CC patients’ highest lymph nodes (2.558 mRNA copies/18S rRNA unit). In (**B**,**D**,**F**), the Kaplan-Meier cumulative survival curves for EpCAM, LGR5 and LGR4 patients are restricted to the CEA(+) plus CEA(int) subgroup of CC patients. In (**G**–**J**), the Kaplan-Meier cumulative survival curves for LGR5 patients are restricted to TNM stage I patients only (**G**), CEA(int) plus CXCL16(+) subgroups of patients (**H**), CXCL17(+) patients only (**I**) and CEA(+) plus CEA(int) plus CXCL17(+) patients subgroups (**J**). The patients were followed for 12 years. Differences in disease-free survival time after surgery between the two groups are given as a ∆-value in months and statistical significance as *p*-values. n = number of patients in the respective group.

**Table 1 ijms-23-00403-t001:** Correlations between mRNA expressions of different markers in the lymph nodes of colon cancer patients.

		CEA		CXCL17		CXCL16		GPR35 V2/3		LGR5		LGR4	
		r	*p*-Value	r	*p*-Value	r	*p*-Value	r	*p*-Value	r	*p*-Value	r	*p*-Value
**EpCAM**	All CC LNs	0.48	<0.0001	0.33	<0.0001	0.63	<0.0001	0.57	<0.0001	0.62	<0.0001	0.67	<0.0001
	TNM Stage I LNs	0.52	<0.0001	0.48	<0.0001	0.64	<0.0001	0.6	<0.0001	0.57	<0.0001	0.65	<0.0001
	TNM Stage II LNs	0.15	0.035	0.18	0.015	0.52	<0.0001	0.44	<0.0001	0.5	<0.0001	0.55	<0.0001
	TNM Stage III LNs	0.69	<0.0001	0.39	0.0002	0.64	<0.0001	0.68	<0.0001	0.72	<0.0001	0.75	<0.0001
	TNM Stage IV LNs	0.86	<0.0001	0.64	0.0001	0.76	<0.0001	0.76	<0.0001	0.88	<0.0001	0.75	<0.0001
**LGR5**	All CC LNs	0.28	<0.0001	0.31	<0.0001	0.44	<0.0001	0.52	<0.0001			0.68	<0.0001
	TNM Stage I LNs	0.18	0.12	0.29	0.02	0.54	<0.0001	0.6	<0.0001			0.8	<0.0001
	TNM Stage II LNs	0.02	0.79	0.18	0.01	0.3	<0.0001	0.42	<0.0001			0.53	<0.0001
	TNM Stage III LNs	0.47	<0.0001	0.39	0.0002	0.46	<0.0001	0.53	<0.0001			0.76	<0.0001
	TNM Stage IV LNs	0.81	<0.0001	0.66	<0.0001	0.61	0.0003	0.79	<0.0001			0.75	<0.0001
**LGR4**	All CC LNs	0.23	<0.0001	0.24	<0.0001	0.69	<0.0001	0.56	<0.0001				
	TNM Stage I LNs	0.21	0.07	0.31	0.0088	0.73	<0.0001	0.67	<0.0001				
	TNM Stage II LNs	0.13	0.08	0.07	0.3255	0.61	<0.0001	0.42	<0.0001				
	TNM Stage III LNs	0.45	<0.0001	0.37	0.0004	0.66	<0.0001	0.63	<0.0001				
	TNM Stage IV LNs	0.58	0.0006	0.26	0.155	0.72	<0.0001	0.58	0.0006				

The correlation coefficients (r) and the *p* values were calculated by two-tailed Spearman’s rank order correlation test.

**Table 2 ijms-23-00403-t002:** Comparative analysis of average survival time after surgery and risk for recurrence of disease of CC patients with EPCAM(-) and EpCAM(+), LGR5(-) and LGR5(+) and LGR4(-) and LGR4(+) lymph nodes.

Patient Group	Category	Number of Patients in Each Group Stratified by the TNM Stage of Colon Cancer					5 Years Follow-Up after Surgery					12 Years Follow-Up after Surgery				
							Disease-Free Survival			Risk for Recurrence		Disease-Free Survival			Risk for Recurrence	
		Stage I	Stage II	Stage III	Stage IV	Total	Average ^a^	Difference	*p*-Value	Hazard Ratio	*p*-Value	Average ^a^	Difference	*p*-Value	Hazard Ratio	*p*-Value
							(months)	(months)		(95% CI) ^b^		(months)	(months)		(95% CI) ^b^	
All CC patients	EpCAM(-) ^c^	18	40	21	2	81	54					118				
	EpCAM(+)	5	12	16	7	40	46	8	0.007	2.5	0.010	84	34	0.022	2.1	0.025
										(1.2–4.9)					(1.1–4.1)	
	LGR5(-) ^d^	19	39	23	2	83	45					120				
	LGR5(+)	4	13	14	7	38	37	8	0.001	3.0	0.002	78	42	0.005	2.5	0.007
										(1.5–5.8)					(1.3–4.8)	
	LGR4(-) ^e^	19	45	25	2	91	54					117				
	LGR4(+)	4	7	12	7	30	43	11	0.002	2.8	0.004	75	42	0.011	2.3	0.014
										(1.4–5.6)					(1.2–4.6)	
CEA(int) plus CEA(+) ^f^ CC patients	EpCAM(-)	10	20	9	2	41	53					119				
	EpCAM(+)	3	10	15	7	35	46	7	0.055	2.2	0.062	83	36	0.056	2.2	0.063
										(0.9–5.0)					(0.9–5.0)	
	LGR5(-)	12	21	12	2	47	54					122				
	LGR5(+)	1	9	12	7	29	43	11	0.009	2.8	0.012	72	50	0.006	3.0	0.008
										(1.3–6.4)					(1.3–6.8)	
	LGR4(-)	11	25	13	2	51	53					118				
	LGR4(+)	2	5	11	7	25	44	9	0.024	2.4	0.029	72	46	0.019	2.5	0.024
										(1.1–5.4)					(1.1–5.7)	

^a^ Mean survival time after surgery for CC patients as calculated by cumulative survival analysis according to Kaplan–Meier. ^b^ Hazard ratio with 95% confidence interval (CI) for CC patients as calculated according to univariate COX regression analysis. ^c^ CC patients divided into two groups EpCAM(-) and EpCAM(+) according to the median of mRNA expression in the highest lymph nodes in the CC patients in TNM stages III and IV (0.07 mRNA copies/18S rRNA unit). ^d^ CC patients divided into two groups LGR5(-) and LGR5(+) according to the median of mRNA expression in lymph nodes in all CC patients in TNM stage IV (0.06 mRNA copies/18S rRNA unit). ^e^ CC patients divided into two groups LGR4(-) and LGR4(+) according to the 75th percentile of mRNA expression values in all CC patients’ highest lymph nodes (2.558 mRNA copies/18S rRNA unit). ^f^ CEA(int) plus CEA(+) group: CC patients where the CEA mRNA levels in the highest lymph node is higher than the highest level of control patients’ lymph nodes, i.e., 0.013 mRNA copies/18S rRNA unit.

**Table 3 ijms-23-00403-t003:** Comparative analysis of average survival time after surgery and risk for recurrence of disease of CC patients with LGR5(-) and LGR5(+) lymph nodes when combined with other markers or restricted to a certain group of CC patients:.

Patient Group	Category	Number of Patients in Each Group Stratified by the TNM Stage of Colon Cancer					5 Years Follow-Up after Surgery					12 Years Follow-Up after Surgery				
							Disease-Free Survival			Risk for Recurrence		Disease-Free Survival			Risk for Recurrence	
		Stage I	Stage II	Stage III	Stage IV	Total	Average ^a^	Difference	*p*-Value	Hazard Ratio	*p*-Value	Average ^a^	Difference	*p*-Value	Hazard Ratio	*p*-Value
							(months)	(months)		(95% CI) ^b^		(months)	(months)		(95% CI) ^b^	
Stage I CC patients	LGR5(-) ^d^	19	0	0	0	19	60					136				
	LGR5(+)	4	0	0	0	4	42	18	0.007	13.3	0.036	82	54	0.033	8.7	0.077
										(1.2–150.5)					(0.8–96.7)	
CEA(int) and CXCL16(+) CC patients (>7.2) ^c^	LGR5(-) ^d^	4	7	2	0	13	57					121				
	LGR5(+)	0	3	3	0	6	46	11	0.039	7.6	0.079	58	63	0.014	10.4	0.044
										(0.8–73.3)					(1.1–102.9)	
CXCL17(+) CC patients (>0.0012) ^e^	LGR5(-) ^d^	3	3	4	0	10	60					126				
	LGR5(+)	3	4	11	4	22	46	14	0.050	6.1	0.087	75	51	0.047	6.2	0.082
										(0.8–47.4)					(0.8–49.1)	
CEA(int) plus CEA(+) and CXCL17(+) CC patients (>0.0012) ^f^	LGR5(-) ^d^	3	3	3	0	9	60					126				
	LGR5(+)	1	3	11	4	19	46	14	0.060	5.8	0.096	73	53	0.043	6.8	0.075
										(0.7–45.9)					(0.8–55.6)	

^a^ Mean survival time after surgery for CC patients as calculated by cumulative survival analysis according to Kaplan-Meier. ^b^ Hazard ratio with 95% confidence interval (CI) for CC patients as calculated according to univariate COX regression analysis. ^c^ CEA(int) and CXCL16(+) group: CC patients with intermediate CEA mRNA levels, that is, 0.013–3.67 copies/18S rRNA unit in the highest lymph nodes and CXCL16 levels higher than the clinical cut-off; >7.2 mRNA copies/18S rRNA unit. ^d^ CC patients divided into two groups LGR5(-) and LGR5(+) according to the median of mRNA expression in lymph nodes in all CC patients in TNM stage IV (0.06 mRNA copies/18S rRNA unit). ^e^ CXCL17(+) group: CC patients with CXCL17 levels higher then the 73rd percentile; >0.0012 mRNA copies/18S rRNA unit in the highest lymph nodes. ^f^ CEA(int) plus CEA(+) and CXCL17(+) group: CC patients where the CEA mRNA levels in the highest lymph node is higher than the highest level of control patients’ lymph nodes, i.e., 0.013 mRNA copies/18S rRNA unit and CXCL17 is higher than 0.0012 mRNA copies/18S rRNA.

## Data Availability

Data used in this study could be provided upon request, only with permission of the authors of the original studies.

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
