# Peer review of "Clinical Significance of Stem Cell Biomarkers EpCAM, LGR5 and LGR4 mRNA Levels in Lymph Nodes of Colon Cancer Patients"

_ijms, 2021, doi:10.3390/ijms23010403_

Round 1
Reviewer 1 Report
The experimental hypothesis, the test method and the obtained results are significant and could represent in the near future a method applicable in the diagnosis for the prognosis of cancers, as well as in the modulation of the treatment according to the risk factors.
Without diminishing the significance of the results presented, it should be noted that cancers have different genetic changes from one patient to another, even if they have similar histological changes. Moreover, in subsequent steps, consideration should be given to establishing experimental groups that differentiate in addition to a high sensitivity (prognosis) and a high specificity in differentiating possible associated pathologies, or possible clinical situations (such as cell lysis not directly related to the existence of metastases but with changes due to inflammation, lymphatic stasis, therapeutic toxicity or over-added infections).
Author Response
We thank the reviewer very much for the positive comments on our manuscript.
We totally agree with the reviewer’s reference to the importance of individual genetic diversity of cancers and of the importance of "non-cancerous factors" affecting the patient. We aim to perform further studies in this area using a significantly larger clinical material.
Reviewer 2 Report
The authors have done a commendable work analyzing mRNA levels of cancer stem cells biomarkers in colon cancer patients to determine survival outcomes after surgery.
The authors have done a in-depth analysis of the biomarkers in cancer cells, colon cancer tumors and colon cancer patients.
The authors have presented their finding in comprehensible graphs and tables using robust statistical analysis.
My only recommendation to the authors is to verify the numbers in section 2.2 line 119 and 120, where the expression levels of EpCAM don't correlate with what is stated in lines 121 and 122 and also don't match the data in Figure 1B
I highly commend the authors for their work and wish them good luck.
Author Response
We thank the reviewer very much for the positive comments on our manuscript.
We believe that the reviewer has misunderstood the description of EpCAM expression levels of stage I through IV CC lymph nodes in section 2.2. The median values in the different stage groups written in the text match the values indicated by the red horizontal lines in Figure 1B. Although the differences between the median values are relatively small, they are indeed significant using Kruskal-Wallis one-way analysis of variance (ANOVA) test followed by the Dunn’s multiple comparison post-hoc test. It should also be noted that the scale in the Y-axis of the graph is logarithmic (log 10).